# DDM-TS: Decoupled Diffusion Models for Time-Series Generation with Explicit Trend–Seasonality Decomposition

## Abstract

Time-series data are crucial for analysis and prediction across various domains, including finance, healthcare, and energy. However, issues of data scarcity, sensitivity concerns, and privacy regulations limit the availability of high-quality datasets, motivating growing research interest in time-series generation. While recent diffusion models have demonstrated stable learning capabilities and high-quality results for time-series generation, existing approaches face challenges in maintaining structural integrity when generating unconditional multivariate sequences over extended periods. This challenge involves preserving both long-term *trends* and periodic *seasonality* simultaneously, as unified frameworks suffer from inherent component interference that diminishes generation quality. To address this challenge, we propose DDM-TS, a novel decoupled diffusion framework that trains independent diffusion models for trend and seasonality components decomposed via STL decomposition. The framework then employs an adaptive gate-based fusion module that provides learnable fusion weights to unify the independently generated components into coherent synthetic time series. By decoupling trend and seasonal processing pathways, DDM-TS effectively alleviates mutual interference while preserving structural characteristics. Comprehensive experiments across benchmark datasets demonstrate that DDM-TS outperforms state-of-the-art baselines across diverse evaluation measures, achieving an average improvement of 33.8% in reconstruction metrics (RMSE, PSNR), while preserving distributional coverage and frequency-domain consistency as demonstrated by t-SNE and spectral analysis metrics. Code is available at `https://anonymous.4open.science/r/DDM-TS-0D44/`.

## 1 Introduction

Time-series information serves as a fundamental resource for analytical processes and predictive modeling across a wide range of domains, including finance (Zhang et al., 2024), healthcare (Patharkar et al., 2024), and energy (Omoyele et al., 2024). However, obtaining high-quality time-series data is often challenging due to data scarcity, privacy concerns, and regulatory restrictions. To address these limitations, time-series generation methods have received increasing research attention, as they can synthesize artificial data while preserving the characteristics of real datasets. These methods enable data augmentation to enhance task performance while addressing data sensitivity and maintaining privacy concerns.

Early approaches for time-series generation primarily employed VAEs (Desai et al., 2021; Acciaio et al., 2024) and GANs (Yoon et al., 2019; Li et al., 2022). However, VAEs often generate excessively smooth outputs that fail to capture variations in time-series data. GANs suffer from training instability and mode collapse, which limit data diversity and hinder their ability to model long-term dependencies. Recently, denoising diffusion probabilistic models (DDPMs) (Ho et al., 2020) have demonstrated remarkable progress in generative modeling. DDPMs mitigate training instabilities and consistently produce high-quality outputs, achieving state-of-the-art performance across various modalities, including images, videos, and text (Dhariwal & Nichol, 2021; Harvey et al., 2022; Yu et al., 2022). Building on this success, diffusion models have been adapted for time-series generation, offering stable training dynamics and the ability to preserve temporal dependencies through a denoising framework (Rasul et al., 2021a; Feng et al., 2024).

Despite these advances, diffusion-based time-series generation still faces limitations. Prior studies have primarily focused on conditional tasks, such as imputation (Tashiro et al., 2021; Alcaraz & Strodthoff, 2022) and forecasting (Kollovieh et al., 2023; Meijer & Chen, 2024), which typically require additional conditioning inputs such as historical observations or target timestamps. While several approaches have explored unconditional generation, these efforts have often been constrained to either univariate settings (Chen et al., 2020; Kong et al., 2020) or short-term sequences (Lim et al., 2023). The extension to multivariate, long-term unconditional generation remains challenging, necessitating careful preservation of temporal structure.

These complexities primarily arise from the heterogeneous temporal structures in real-world time series. Such series typically exhibit two essential components: *trend*, representing long-term directional changes, and *seasonality*, encompassing periodic fluctuations that recur at regular intervals, as decomposed through seasonal-trend decomposition using Loess (STL) (Cleveland et al., 1990). The limitation of conventional diffusion approaches lies in their uniform application of noise across temporal dimensions, which inadequately accounts for the distinct characteristics of trends and seasonality. This uniform noise application gradually degrades temporal dependencies, disrupting the structural patterns necessary for high-quality synthetic data. While some prior research (Yuan & Qiao, 2024a) has attempted to model trends and seasonality within a single model, we observed that this approach still operates within shared parameter spaces, revealing opportunities for better utilization of component-specific characteristics through specialized architectural designs.

To address these challenges, we introduce DDM-TS, a novel decoupled diffusion architecture that explicitly separates trend and seasonality into independent diffusion models designed to capture their distinct temporal characteristics. The trend pathway applies smoothness filtering operations, while the seasonality pathway incorporates FFT-based spectral consistency terms, with each pathway specialized according to its respective temporal characteristics. This architectural design reduces the component interference that can occur in unified processing frameworks. Our adaptive gate-based fusion module integrates these independent components through learnable weighting parameters that adapt to varying temporal characteristics. This component-aware approach enables better preservation of temporal structure compared to a uniform framework.

In summary, the main contributions of this study are as follows:

- We propose a decoupled diffusion framework that employs independent models for trend and seasonality components, addressing the limitations of unified frameworks in preserving distinct temporal characteristics.
- We design a gate-based fusion module with weighting parameters that integrates independently generated components while maintaining component-specific structural properties.

## 2 RELATED WORKS

**Generative Time-Series Models.** Deep learning–based generative models have been utilized for time-series synthesis across multiple domains. Variants of VAEs, such as TimeVAE (Desai et al., 2021) and KoVAE (Naiman et al., 2024), aimed to enhance the quality of time-series generation. GAN-based approaches were also widely studied, with models such as C-RNN-GAN (Mogren, 2016), RCGAN (Esteban et al., 2017), and TimeGAN (Yoon et al., 2019) designed to capture temporal dependencies. More recent works, including RTSGAN (Pei et al., 2021) and PSA-GAN (Paul et al., 2021), investigated long-term sequence generation through progressive training and self-attention mechanisms. Other alternatives included energy-based generative approaches, such as ScoreGrad (Yan et al., 2021), flow-based methods, including Fourier Flows (Alaa et al., 2021), and implicit neural representations, like SIREN (Sitzmann et al., 2020). While these approaches improved time-series generation, these methods struggle to maintain detailed temporal patterns when generating unconditional data, especially for datasets with multivariate and long time sequences.

**Diffusion-Based Time-Series Generation.** Diffusion models showed significant progress in improving several time-series generation tasks. TimeGrad (Rasul et al., 2021b) used DDPMs with RNNs for autoregressive conditional forecasting. CSDI (Tashiro et al., 2021) introduced a conditional score-based diffusion model that employs self-supervised techniques with masking for time-series imputation. SSSD (Lopez Alcaraz & Strodthoff, 2022) proposed structured state space models within diffusion frameworks for imputation and forecasting to capture long-range dependencies. TimeLDM (Qian et al., 2024) achieved improvements in long-term generation by integrating a VAE

with a latent diffusion model to utilize the VAE's latent space to improve scalability for extended sequences. PaD-TS (Li et al., 2025) addressed unconditional multivariate generation through a population-aware objective and dual-channel encoder to maintain population-level properties such as value distributions and cross-correlation. Although these advances expanded the capabilities of diffusion-based time-series generation, current approaches do not fully leverage the decomposed nature of temporal patterns, thereby missing opportunities to utilize component-specific modeling for trends and seasonality.

**Trend–Seasonality Decomposition for Time-Series Modeling.** Prior studies have explored integrating decomposition principles such as STR (Dokumentov & Hyndman, 2020) and RobustSTL (Wen et al., 2019) for time-series forecasting and imputation. SSDNet (Lin et al., 2021) combined a transformer with a state-space decomposition architecture to separately model temporal components, thereby enhancing interpretability and predictive accuracy. SpectraNet (Challu et al., 2022) utilized a spectral decomposition of the latent space to perform multivariate analysis. This approach showed robust performance under distribution shifts and missing data. Recent research has explored the integration of temporal decomposition principles with diffusion models to address complex time-series modeling. Mr-Diff (Shen et al., 2024) introduced a multi-resolution diffusion framework that separated time series into hierarchical temporal scales for forecasting. It employed a coarse-to-fine process that first established overall temporal structure before progressively adding finer details. FALDA (Wang et al., 2025) introduced a frequency-aware conditional diffusion framework for forecasting, utilizing Fourier-based decomposition to generate temporal components across different frequencies. Diffusion-TS (Yuan & Qiao, 2024b) employs a disentangled temporal representation and reconstruction-based sampling for unconditional time-series generation. While these methods demonstrated progress in incorporating temporal decomposition, we identified opportunities for enhancement: they employ a single unified model to process all decomposed components (trends and seasonality). This approach revealed potential for improvement in how the unique characteristics of each component are utilized, suggesting that decoupled architectures could better capture the distinct temporal behaviors of each component.

## 3 DDM-TS: Decoupled Diffusion Models for Time Series

We propose DDM-TS, a decoupled diffusion architecture that employs separate DDPMs for trend and seasonality synthesis. Our method is designed around the principle that trends exhibit gradual long-term changes while seasonality displays repetitive cycles. These different temporal behaviors require separate modeling strategies to capture them effectively.

Based on this insight, the framework operates through a three-stage process: First, input data undergoes STL decomposition (Cleveland et al., 1990) using the operator $\mathrm{STL}(x; m, \psi)$ to extract trend and seasonality components. Second, each component is processed by a decoupled diffusion model. The trend model utilizes smoothness filtering with learnable thresholds to preserve smooth temporal transitions, while the seasonality model employs Fourier layer adjustments to reflect periodic patterns. Third, a gating mechanism dynamically weights and reconstructs the final output to balance component contributions based on characteristics of each component. This decoupled design enables each diffusion model to focus exclusively on component-specific temporal dynamics. The architecture of DDM-TS is illustrated in Figure 1.

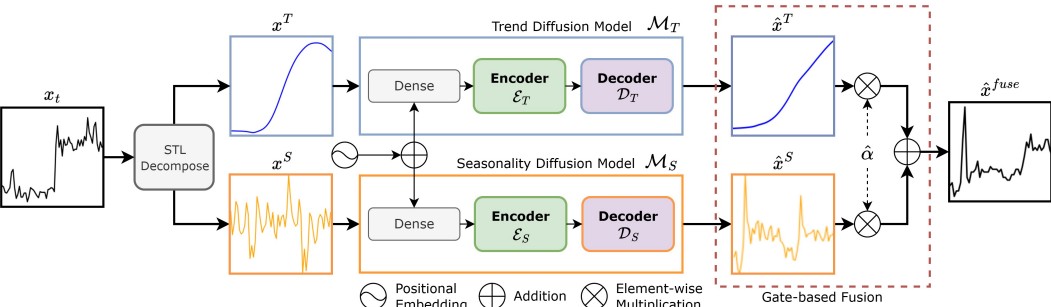

Figure 1: DDM-TS model architecture

### 3.1 COMPONENT-SPECIFIC ARCHITECTURAL DESIGN

In the proposed framework, the input time series $x$ is decomposed using the STL method, where $x^T$ represents the trend component, $x^S$ denotes the seasonal component, and $x^R$ means the residual component. The residual component $x^R$ is integrated into the seasonality pathway as $x^S \leftarrow x^S + x^R$, as residuals contain irregular variations that are essential for generating realistic time series following the approach established in (Yuan & Qiao, 2024b).

$$(x^T, x^S, x^R) \leftarrow \text{STL}(x; m, \psi). \tag{1}$$

These decomposed components then undergo the forward diffusion process, where Gaussian noise is progressively added over $t$ timesteps to obtain noisy sequences $x_t^T$ and $x_t^S$. The proposed framework employs a dual-pathway denoising architecture comprising independent Transformer encoder–decoder networks for trend and seasonality modeling. The mathematical formulations for reconstructing the denoised components $\hat{x}_t^T$ and $\hat{x}_t^S$ are expressed as follows:

$$\hat{x}_t^T = \mathcal{D}_T\big(\mathcal{E}_T(x_t^T) + \mathcal{P}(t) + \mathcal{P}_{\text{trend}}(\alpha_T(t))\big), \tag{2}$$

$$\hat{x}_t^S = \mathcal{D}_S\big(\mathcal{E}_S(x_t^S) + \mathcal{P}(t) + \mathcal{P}_{\text{seasonal}}(\omega)\big), \tag{3}$$

where $\mathcal{E}_T(\cdot)$ and $\mathcal{E}_S(\cdot)$ denote the trend and seasonality encoders, while $\mathcal{D}_T(\cdot)$ and $\mathcal{D}_S(\cdot)$ represent their corresponding decoders. The term $\mathcal{P}(t)$ provides timestep embedding, while the trend embedding $\mathcal{P}_{\text{trend}}(\alpha_T(t))$ incorporates the fusion parameter $\alpha_T(t)$ to guide trend reconstruction, where $\alpha_T(t)$ controls the relative weighting of trend components, as detailed in Section 3.2. The seasonality positional encoding $\mathcal{P}_{\text{seasonal}}(\omega)$ encodes cyclical temporal relationships using sinusoidal functions with frequency $\omega$.

Both independent encoders employ the same multi-layer architecture that processes input sequences through Transformer blocks:

$$\mathcal{E}_{T/S}(x) = \sum_{k=1}^{K} \Big\{ x + \mathcal{N}_k\big(\text{Attn}_k(\text{FFN}_k(x)), t\big) \Big\}, \tag{4}$$

where $\text{FFN}_k(\cdot)$ corresponds to feed-forward networks applied to input features, $\text{Attn}_k(x)$ denotes the multi-head self-attention operation at layer $k$, and $\mathcal{N}_k(x, t)$ represents timestep-conditioned normalization that adapts to current noise levels. This identical encoder architecture ensures consistent feature extraction across both pathways, while component-specific specialization is achieved through the distinct decoder architectures.

The trend decoder $\mathcal{D}_T$ is designed to capture long-term temporal dependencies through convolution with the smoothing operation:

$$\mathcal{D}_T(x) = \sum_{k=1}^{K} \Big\{ \mathcal{C}_k(x) + \mathcal{C}_k\big(\mathcal{F}_k^{\text{trend}}(x) \odot \text{Proj}_k(x)\big) \Big\}, \tag{5}$$

where $\mathcal{C}_k(x)$ denotes the convolution operation at layer $k$ for trend extraction, and $\text{Proj}_k(x)$ projects the input features into a latent space. $\mathcal{F}_k^{\text{trend}}(x)$ represents a trend-specific filtering operation defined as:

$$\mathcal{F}_k^{\text{trend}}(x) = \frac{1}{1 + \exp\big(|\nabla x| - \tau_k^*\big)} \cdot (\mathcal{W}_k^{\text{smooth}} \circledast x + b_k), \tag{6}$$

where $\mathcal{W}_k^{\text{smooth}}$ and $b_k$ represent smoothness-promoting weight matrices and bias terms, and $\tau_k^*$ is a learnable threshold parameter that determines the gradient magnitude boundary for the filtering operation. The weighting function controls the transition to facilitate smoother gradients (when $|\nabla x| < \tau_k^*$) or suppress steeper gradients (when $|\nabla x| > \tau_k^*$). These components implicitly modulate the contribution of features with varying smoothness levels, enabling each layer to adopt distinct filtering behaviors. The trend-specific filtering operation retains long-term temporal smoothness while suppressing excessive gradients, thereby facilitating robust trend extraction across extended time horizons.

The seasonality decoder $\mathcal{D}_S$ employs frequency-domain processing to capture periodic patterns:

$$\mathcal{D}_S(x) = \sum_{k=1}^{K} \Big\{ \mathcal{A}_k \cdot \mathcal{F}^{-1}\big(\mathcal{F}(\xi_k(x)) \cdot e^{i\phi_k}\big) + \xi_k(x) \Big\}, \tag{7}$$

where $\mathcal{F}(\cdot)$ and $\mathcal{F}^{-1}(\cdot)$ denote FFT and inverse FFT operations, $\mathcal{A}_k$ represents scaling factors, $\phi_k$ denotes phase shifts for aligning periodic patterns, and $\xi_k(x)$ performs layer-specific feature encoding with learnable parameters. $\mathcal{D}_S$ transforms features to the frequency domain, applies phase adjustments to synchronize seasonal cycles, and converts back to the time domain with adaptive scaling. This process enables the seasonality model to effectively identify and manipulate periodic patterns, particularly when multiple seasonal cycles are present within multivariate sequences.

## 3.2 Gate-based Fusion and Loss Function

The independent components generated from the decoupled diffusion models are integrated through a temporal-aware fusion module. In the fusion process, the learnable parameters $\alpha_T(t)$ and $\alpha_S(t)$ are designed to dynamically capture the importance of trend and seasonality components through adaptive weighting that reflects their varying contributions across different temporal contexts. This adaptive weighting mechanism enables the model to automatically emphasize the trend components during long-term directional transitions and the seasonal components during periodic patterns and recurring cycles. The parameters exhibit frequency-dependent patterns where $\alpha_T(t)$ increases in low-frequency regions while $\alpha_S(t)$ responds in high-frequency regions. The fused output is expressed as follows, where $\odot$ denotes element-wise multiplication.

$$\hat{x}_t^{\text{fuse}} = \alpha_T(t) \odot \hat{x}_t^T + \alpha_S(t) \odot \hat{x}_t^S. \tag{8}$$

For the trend component, the loss function is as follows. This loss incorporates no additional regularization terms because of leveraging the architectural design of the trend decoder to promote smoothness:

$$\mathcal{L}_T = \mathbb{E}_{x^T, \epsilon_T, t} \left[ \|\epsilon_T - \epsilon_{\theta_T}(x_t^T, t)\|^2 \right]. \tag{9}$$

For the seasonal component, the loss function incorporates periodic consistency and is as follows. The periodicity penalty term $\lambda_{\text{periodic}} \|\mathcal{F}(\hat{x}_t^S) - \mathcal{F}(x_t^S)\|^2$ computes separate losses on the real and imaginary components of the Fourier transform to capture comprehensive frequency-domain information:

$$\mathcal{L}_S = \mathbb{E}_{x^S, \epsilon_S, t} \left[ \|\epsilon_S - \epsilon_{\theta_S}(x_t^S, t)\|^2 \right] + \lambda_{\text{periodic}} \|\mathcal{F}(\hat{x}_t^S) - \mathcal{F}(x_t^S)\|^2. \tag{10}$$

To ensure optimal fusion quality and maintain fidelity to the original time series structure, we introduce an additional fusion loss that measures the reconstruction accuracy of the combined output, where $x_t$ represents the original time series values at timestep $t$ and $\hat{x}_t^{\text{fuse}}$ denotes the corresponding fused reconstruction:

$$\mathcal{L}_{\text{fuse}} = \mathbb{E}_{x, t} \left[ \|\hat{x}_t^{\text{fuse}} - x_t\|^2 \right]. \tag{11}$$

The overall training objective combines all loss components to optimize both individual component quality and their integration:

$$\mathcal{L}_{\text{total}} = \mathcal{L}_T + \mathcal{L}_S + \lambda_{\text{fuse}} \mathcal{L}_{\text{fuse}}, \tag{12}$$

where $\lambda_{\text{fuse}}$ is a hyperparameter that controls the contribution of the fusion loss, ensuring effective component integration while preserving the overall time series characteristics.

---

**Algorithm 1** Training DDM-TS with decoupled diffusion models

---

**Require:** Input time series $x$, trend and seasonality diffusion models, fusion parameters
 1: Decompose input using STL decomposition
 2: Integrate residual into seasonality pathway
 3: Initialize model parameters and fusion weights
 4: **for** each training iteration **do**
 5:     Sample timestep $t$ and noise $\epsilon_T, \epsilon_S \sim \mathcal{N}(0, I)$
 6:     Apply forward diffusion to both components independently
 7:     Encode trend and seasonality with component-specific embeddings
 8:     Decode to reconstruct components
 9:     Apply adaptive fusion to combine components
10:     Compute losses and update all parameters via gradient descent
11: **end for**
12: **return** trained models and fusion parameters

---

Table 1: Performance comparison across multiple time-series datasets using sequences of length 24. Best results are shown in **bold red**, second-best results in **bold blue**.

| Dataset | Method | RMSE | PSNR | COS | MMD | MDD | ACD | SD | KD | ED | DTW |
|---|---|---|---|---|---|---|---|---|---|---|---|
| Stock | TimeVAE | 0.2996 | 7.6704 | 0.9410 | 0.2336 | 0.0436 | 0.1451 | **0.3543** | **2.1447** | 3.4134 | 17.1984 |
| | TimeGAN | 0.2663 | 8.0411 | 0.9567 | 0.2239 | 0.0409 | 0.0852 | 0.4866 | 2.5348 | 3.3494 | 17.2150 |
| | Diffusion-TS | **0.2327** | **10.1650** | **0.9733** | **0.1821** | 0.0337 | **0.0458** | 0.3690 | **1.5925** | **2.8464** | **12.8309** |
| | PaD-TS | 0.2547 | 9.1886 | 0.9708 | 0.2052 | **0.0128** | **0.0613** | **0.3687** | 4.3857 | 3.1044 | 16.5363 |
| | DDM-TS | **0.0590** | **12.7437** | **0.9768** | **0.0121** | **0.0141** | 0.1577 | 0.4704 | 2.3967 | **0.7498** | **3.4033** |
| Energy | TimeVAE | **0.2413** | **6.6173** | 0.9602 | **0.1238** | 0.0507 | 0.0747 | **0.1408** | **0.6772** | **6.2968** | **30.4819** |
| | TimeGAN | 0.2885 | 5.2175 | 0.9483 | 0.1663 | 0.0550 | 0.1035 | 0.2532 | 0.9920 | 7.3678 | 34.7307 |
| | Diffusion-TS | 0.2512 | 6.1968 | 0.9613 | 0.1344 | **0.0113** | **0.0425** | 0.1659 | 1.2871 | 6.5878 | 31.5602 |
| | PaD-TS | 0.2490 | 6.2479 | **0.9621** | 0.1327 | **0.0079** | **0.0237** | **0.1095** | 0.8748 | 6.5476 | 31.7835 |
| | DDM-TS | **0.1507** | **6.5847** | **0.9805** | **0.0483** | 0.0207 | 0.2083 | 0.2731 | **0.7458** | **3.9890** | **18.3240** |
| ETTh | TimeVAE | 0.2114 | 7.7537 | 0.9803 | 0.0953 | 0.0556 | 0.0717 | 0.4912 | 2.3484 | 2.7618 | 13.3184 |
| | TimeGAN | 0.2454 | 6.3351 | 0.9707 | 0.1244 | 0.0721 | 0.0337 | **0.1421** | 0.8797 | 3.2067 | 14.8134 |
| | Diffusion-TS | 0.1832 | 8.6217 | 0.9829 | 0.0791 | **0.0115** | **0.0231** | 0.1788 | **0.7849** | 2.5067 | 11.3236 |
| | PaD-TS | **0.1789** | **8.9418** | **0.9838** | **0.0740** | **0.0161** | **0.0312** | **0.1415** | 1.1233 | **2.4188** | **11.0997** |
| | DDM-TS | **0.1242** | **12.0503** | **0.9911** | **0.0354** | 0.0169 | 0.1222 | 0.4209 | **0.5783** | **1.6983** | **7.7532** |

## 4 EXPERIMENTAL RESULTS

We evaluate DDM-TS performance through a comprehensive comparison against four time-series generation models: TimeVAE (Desai et al., 2021), TimeGAN (Yoon et al., 2019), Diffusion-TS (Yuan & Qiao, 2024b), and PaD-TS (Zhou et al., 2023), utilized as baseline methods in recent studies (Li et al., 2025; Yuan & Qiao, 2024b). Experiments are conducted across three benchmark datasets: Stock, Energy, and ETTh datasets. Detailed descriptions of datasets and evaluation metrics are provided in Appendix C.

### 4.1 UNCONDITIONAL TIME SERIES GENERATION

Table 1 presents unconditional time-series generation results with sequence length 24, where DDM-TS achieves the best performance in 18 out of 30 cases. These results demonstrate substantial improvements in reconstruction metrics, including an approximately 75% reduction in RMSE on the Stock dataset and enhanced PSNR scores across all datasets. DDM-TS excels in reconstruction and distance-based measures (ED, DTW) while maintaining strong temporal dependencies (COS). DDM-TS excels in reconstruction and distance-based measures (ED, DTW), while showing mixed performance in distributional metrics. Notably, DDM-TS underperforms in frequency-domain (SD) and autocorrelation-based (ACD) metrics, as well as marginal distribution differences (MDD), suggesting limitations in preserving fine-grained statistical properties. This pattern suggests that the decoupled architecture, although effective in capturing major structural patterns through separate trend and seasonality modeling, may inadvertently reduce the spectral and autocorrelation signatures. These results indicate that while DDM-TS excels in reconstruction accuracy, the method faces inherent trade-offs between structural coherence and the preservation of statistical complexity.

**Data distribution analysis.** Figure 2 presents visualizations for distributional analysis using t-distributed stochastic neighbor embedding (t-SNE) (van der Maaten & Hinton, 2008) and kernel density estimation (KDE) (Silverman, 1986) to provide qualitative insights into the distributional similarity and diversity of generated sequences. The results of t-SNE in the top row reveal that

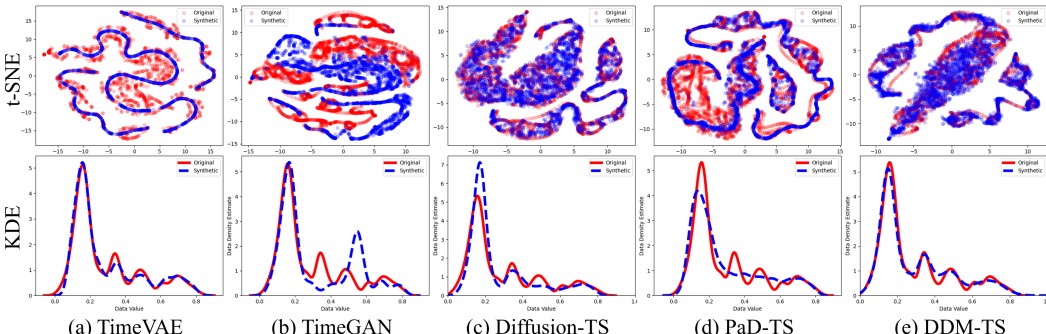

| (a) TimeVAE | (b) TimeGAN | (c) Diffusion-TS | (d) PaD-TS | (e) DDM-TS |

Figure 2: Distributional fidelity comparison across various time-series generation methods

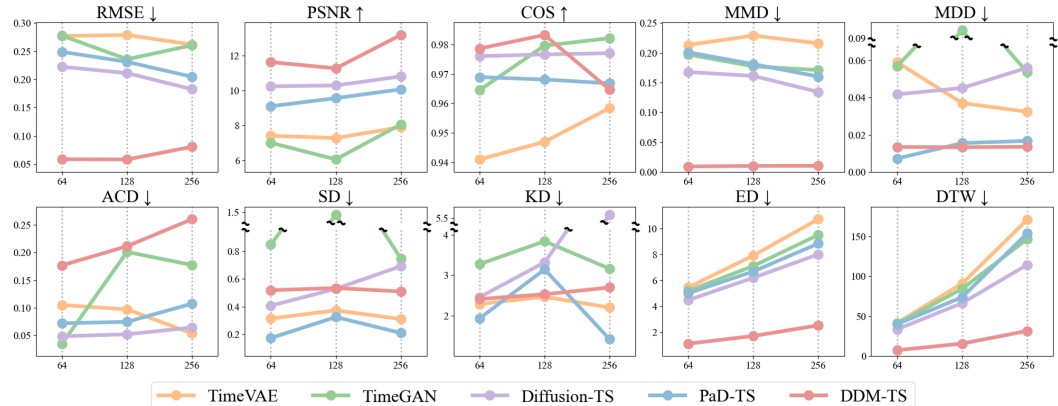

Figure 3: Performance comparison of DDM-TS and baseline methods on stock datasets with extended sequence lengths

DDM-TS, alongside Diffusion-TS, generates diverse synthetic data that closely aligns with the original data distribution, as demonstrated by the widespread and overlapping scatter patterns. Conversely, TimeGAN exhibits distributional misalignment, where the generated samples fail to capture the original data patterns accurately. TimeVAE and PaD-TS demonstrate constrained generation capabilities, producing samples concentrated within limited ranges. The KDE plots in the bottom row further confirm that DDM-TS exhibits close alignment between the original and synthetic data distributions, compared to the baseline methods. These results indicate that DDM-TS effectively preserves the distributional diversity and fidelity of the original data.

**Longer sequence generation.** Figure 3 presents a comprehensive evaluation of DDM-TS against baselines across longer sequence lengths (64, 128, 256) on stock time series data. DDM-TS achieves strong performance in reconstruction metrics (RMSE, PSNR) and distributional consistency (MMD), while maintaining competitive results in distance-based measures (ED, DTW). These results demonstrate DDM-TS's enhanced capability to preserve distinct temporal characteristics. Additionally, MDD achieves robust outcomes similar to PaD-TS across all durations, suggesting that DDM-TS maintains stable distributional fidelity regardless of sequence length. However, the evaluation reveals mixed results in temporal correlation measures. COS excels at shorter durations but deteriorates at sequence lengths of 256, revealing difficulty with correlation consistency over extended sequences. Similarly, DDM-TS exhibits moderate outcomes in diversity measures (SD, KD), but consistently poor ACD scores across all sequence durations. This result underscores the challenges in maintaining appropriate diversity characteristics. In summary, these results demonstrate that DDM-TS offers advantages in reconstruction quality and distributional fidelity, while facing limitations in preserving complex temporal correlation patterns and diversity characteristics.

Table 2: Population-level distributional properties across multiple time-series datasets of length 24. The best performance is indicated in **bold**.

| Dataset | Method | VDS | FDDS | DA | PD |
|---------|--------|-----|------|----|----|
| Stock | TimeVAE | 0.1068 | 2.9361 | 0.3577 | 0.1122 |
| | TimeGAN | 0.0404 | 0.0773 | 0.1453 | 0.0383 |
| | Diffusion-TS | 0.0279 | 0.1990 | 0.0689 | **0.0369** |
| | PaD-TS | **0.0031** | 0.0594 | 0.0782 | 0.0373 |
| | DDM-TS | 0.0051 | **0.0260** | **0.0574** | 0.0373 |
| Energy | TimeVAE | 0.2270 | 0.6011 | 0.4986 | 0.3283 |
| | TimeGAN | 0.0177 | 0.4702 | 0.4974 | 0.3079 |
| | Diffusion-TS | 0.0063 | 0.1979 | 0.1210 | **0.2509** |
| | PaD-TS | **0.0020** | **0.0444** | 0.0924 | 0.2517 |
| | DDM-TS | 0.0062 | 0.0892 | 0.3257 | 0.2516 |
| ETTh | TimeVAE | 0.3191 | 0.4529 | 0.4508 | 0.1344 |
| | TimeGAN | **0.0040** | 0.1527 | 0.1322 | 0.1293 |
| | Diffusion-TS | 0.0061 | 0.0726 | 0.0774 | 0.1272 |
| | PaD-TS | 0.0115 | 0.1335 | 0.1053 | 0.1271 |
| | DDM-TS | 0.0061 | **0.0239** | **0.0657** | **0.1250** |

**Population-level properties.** Table 2 presents the results for population-level property preservation. Detailed descriptions of the metrics are provided in Appendix C. On the Stock dataset, DDM-TS achieves competitive performance across population-level metrics, with particularly strong FDDS and DA scores that rank among the top methods while maintaining reasonable VDS and PD performance compared to baselines. These strong FDDS and DA results suggest that DDM-TS successfully maintains the covariance structures and statistical properties of the original population. The Energy dataset demonstrates moderate DDM-TS performance across VDS, FDDS, and PD, though with relatively weaker DA performance. The weaker DA performance suggests that the generated sequences may exhibit subtle artifacts in temporal correlation patterns, making them

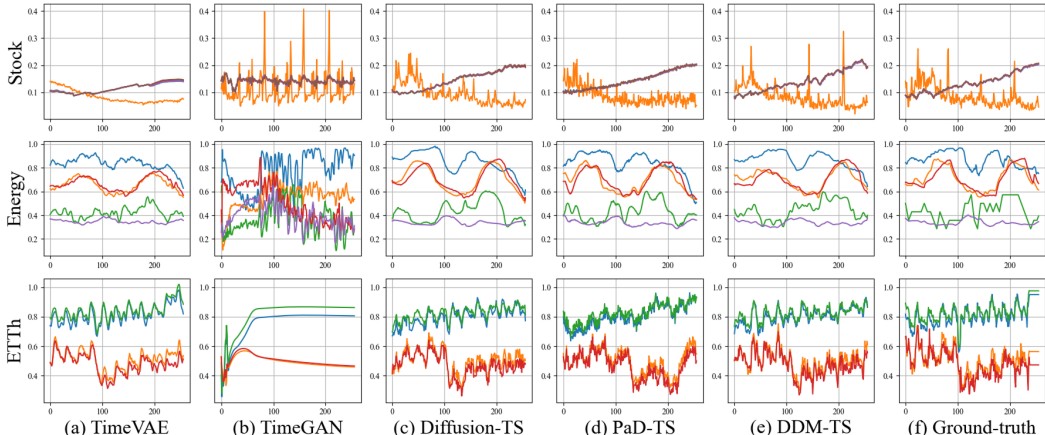

Figure 4: Qualitative comparison of time series generated by DDM-TS and baseline methods for sequence length 256

distinguishable from the original data. The ETTh dataset demonstrates outstanding DDM-TS performance, yielding the best results across most evaluation metrics. Overall, DDM-TS demonstrates that enhanced reconstruction accuracy can be achieved while maintaining population-level preservation performance comparable to state-of-the-art methods.

**Qualitative comparison:** Figure 4 presents qualitative comparisons of time series generated by different methods. TimeVAE produces simplified waveforms, particularly evident in the Stock and Energy datasets, while TimeGAN creates patterns with temporal artifacts and unrealistic fluctuations. Diffusion-TS and PaD-TS demonstrate moderate fidelity but exhibit specific waveform limitations. In the Stock data, both methods show amplitude deviations and smoother waveforms that miss critical volatility spikes. For ETTh sequences, both approaches exhibit gradual drift from the ground-truth characteristics. DDM-TS successfully captures volatility patterns in Stock data and maintains long-term dependencies with temporal coherence in ETTh sequences. While DDM-TS generally achieves superior reconstruction quality, it occasionally exhibits smoother waveforms in Energy data that deviate from expected periodic behavior compared to PaD-TS. In conclusion, these results demonstrate that DDM-TS achieves prominent visual fidelity in most scenarios, capturing dataset-specific characteristics, with limitations observed in certain patterns of the Energy dataset.

## 4.2 ABLATION STUDY

Table 3 shows an ablation study to evaluate the effectiveness of each component in our decoupled diffusion framework. We examine two key variants by removing core components: (1) without the trend module, and (2) without the seasonality module. RMSE and PSNR show the most severe degradation when the trend module is removed, confirming that the trend module is crucial for pointwise accuracy. MMD, MDD, COS, and DTW exhibit more balanced performance degradation when each component is removed. This balanced response occurs because these metrics evaluate distributional consistency and temporal alignment properties that are influenced by both the trend and the seasonality module. ACD, SD, and KD exhibit greater sensitivity to the removal of the

Table 3: Results of ablation study on multiple time-series datasets with 24 length. The best performance is indicated in **bold**.

| Dataset | Method | RMSE | PSNR | COS | MMD | MDD | ACD | SD | KD | ED | DTW |
|---------|--------|------|------|-----|-----|-----|-----|-----|-----|-----|-----|
| Stock | w/o Trend | 0.3110 | 4.2941 | 0.9472 | 0.2012 | 0.2856 | 0.1581 | 0.6776 | 7.6262 | 3.8087 | 18.6823 |
| | w/o Seasonality | 0.2825 | 7.4338 | 0.9478 | 0.1355 | 0.1123 | **0.0583** | **0.3603** | 2.7807 | 2.7588 | 12.7156 |
| | DDM-TS | **0.0590** | **12.7437** | **0.9768** | **0.0121** | **0.0141** | 0.1577 | 0.4704 | **2.3967** | **0.7498** | **3.4033** |
| Energy | w/o Trend | 0.2482 | 5.9106 | 0.9415 | 0.1285 | 0.1669 | 0.0917 | 0.6353 | 2.0788 | 6.6212 | 31.9030 |
| | w/o Seasonality | 0.2503 | 6.2210 | 0.9612 | 0.1339 | **0.0157** | **0.0361** | **0.2614** | 3.2201 | 6.5725 | 31.4547 |
| | DDM-TS | **0.1507** | **6.5847** | **0.9805** | **0.0483** | 0.0207 | 0.2083 | 0.2731 | **0.7458** | **3.9890** | **18.3240** |
| ETTh | w/o Trend | 0.1956 | 7.5162 | 0.9795 | 0.0914 | 0.1117 | 0.0740 | 0.8413 | 2.4560 | 2.7666 | 12.2541 |
| | w/o Seasonality | 0.1830 | 8.6400 | 0.9830 | 0.0787 | **0.0124** | **0.0229** | **0.1657** | 0.8346 | 2.5012 | 11.5111 |
| | DDM-TS | **0.1242** | **12.0503** | **0.9911** | **0.0354** | 0.0169 | 0.1222 | 0.4209 | **0.5783** | **1.6983** | **7.7532** |

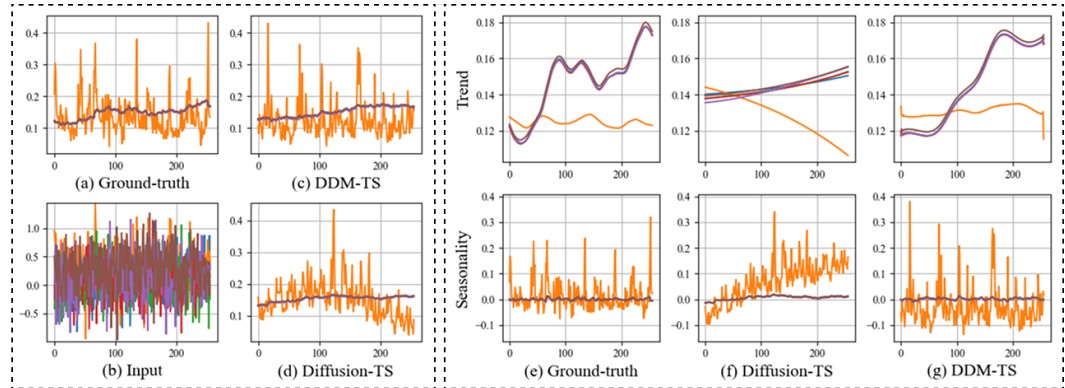

Figure 5: Performance comparison showing (left) noise-to-clean signal reconstruction capabilities, and (right) trend-seasonality decomposition quality

seasonality module, as these diversity and autocorrelation metrics are particularly dependent on capturing temporal variations. On the Stock dataset, removing the trend module results in the most severe performance drops in most metrics, indicating that the Stock dataset relies heavily on smooth trend modeling for accurate generation. The Energy and ETTh datasets demonstrate more evenly distributed contributions from both seasonal and trend components compared to the Stock dataset. The results of full DDM-TS exhibit the best performance, validating that trend and seasonality components are essential for achieving optimal time-series generation quality.

### 4.3 INTERPRETABILITY RESULTS

Figure 5 presents a comprehensive analysis of DDM-TS's reconstruction and decomposition capabilities compared to Diffusion-TS, which also incorporates trend and seasonality modeling, through two distinct evaluation phases.

**Reconstruction:** DDM-TS exhibits enhanced performance in restoring original signal characteristics, thereby preserving the accuracy of both amplitude and fluctuation patterns. The reconstructed signals demonstrate temporal smoothness and maintain periodic oscillations that closely match the ground truth patterns observed in the original data. Conversely, Diffusion-TS struggles to capture the gradual transitions and fluctuation patterns inherent in the original data. This results in the loss of critical signal details essential for representing time series.

**Decomposition:** DDM-TS demonstrates higher decomposition quality in accurately separating trend and seasonality components compared to Diffusion-TS. The trend component exhibits a greater ability to capture complex, non-linear temporal changes compared to the Diffusion-TS model. The seasonality component exhibits ideal periodic behavior, fluctuating around the origin without unwanted drift or long-term movements. In contrast, Diffusion-TS exhibits less precise decomposition performance. The trend component becomes too smooth and loses details, while the seasonality components are prone to unwanted drift.

These comparative results demonstrate that DDM-TS achieves enhanced fidelity in both signal reconstruction and component decomposition by maintaining temporal coherence and preserving distinct trend-seasonality characteristics, without introducing drift artifacts or detail loss.

### 5 CONCLUSION

This research introduces DDM-TS, a novel decoupled diffusion architecture that trains independent diffusion models for trend and seasonality generation. The method applies STL to separate the input time series into trend and seasonal components, which are then processed with decoupled component-specific architectural designs. A gate-based fusion module then adaptively integrates the trend and seasonal components based on temporal context. Experimental results show that DDM-TS outperforms baseline models in most cases, demonstrating improved performance in preserving both distributional consistency and structural properties. DDM-TS presents a novel approach to generating synthetic time series, offering significant potential for future applications in data augmentation, simulation, and forecasting.

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

## A  DIFFUSION MODELS

Denoising diffusion probabilistic models (DDPMs) (Ho et al., 2020; Dhariwal & Nichol, 2021) are probabilistic generative frameworks that employ a two-stage process: a forward process that corrupts data by adding Gaussian noise, and a reverse process that reconstructs clean data through iterative denoising. The forward process transforms a given data sample $x^0$ by incrementally adding Gaussian noise $\epsilon$ according to a fixed variance schedule $\{\beta_t\}_{t=1}^T$ with $t$ uniformly sampled from $\{1, ..., T\}$:

$$q(x^{1:T} \mid x^0) = \prod_{t=1}^{T} q(x^t \mid x^{t-1}), \quad q(x^t \mid x^{t-1}) = \mathcal{N}(\sqrt{1-\beta_t}x^{t-1}, \beta_t\mathbf{I}). \tag{13}$$

The reverse process initiates from pure Gaussian noise $x^T \sim \mathcal{N}(0, \mathbf{I})$ and progressively removes noise through a neural network to recover the original data distribution:

$$p_\theta(x^{0:T}) = p(x^T) \prod_{t=1}^{T} p_\theta(x^{t-1} \mid x^t), \quad p_\theta(x^{t-1} \mid x^t) = \mathcal{N}(\mu_\theta, \Sigma_\theta). \tag{14}$$

Here, the mean $\mu_\theta$ and the variance $\Sigma_\theta$ are learnable parameters. $\Sigma_\theta$ is typically simplified as $\Sigma_\theta = \sigma_t^2\mathbf{I}$. The parameter $\mu_\theta$ can then be expressed as

$$\tilde{\mu}^t(x^t, x^0) = \frac{\sqrt{\bar{\alpha}^{t-1}}\beta^t}{1 - \bar{\alpha}^t}x^0 + \frac{\sqrt{\alpha^t}(1 - \bar{\alpha}^{t-1})}{1 - \bar{\alpha}^t}x^t \tag{15}$$

where $\alpha_t = 1 - \beta_t$, $\bar{\alpha}_t = \prod_{s=1}^t \alpha_s$. The training process optimizes model parameters through a loss function that quantifies the discrepancy between the actual noise $\epsilon$ added during the forward diffusion process and the network's denoised prediction $\epsilon_\theta(x^t, t)$:

$$\mathcal{L}(\theta) = \mathbb{E}_{x^0, \epsilon, t} \left[ \|\epsilon - \epsilon_\theta(x_t, t)\|^2 \right]. \tag{16}$$

## B  IMPLEMENTATION DETAILS

DDM-TS architecture implements dual encoder–decoder Transformer frameworks with distinct configurations for modeling trends and seasonality. The input features of dimension $d$ are transformed into model embedding dimension $d_{\text{model}}$ through a convolutional MLP module. The architecture employs dataset-specific configurations with varying complexity: the Energy dataset utilizes 28 input features with 96 embedding dimensions, 4 encoder layers, and 3 decoder layers; the Stock dataset employs 6 input features with 64 embedding dimensions, 2 encoder layers, and 2 decoder layers; and the ETTh dataset uses seven input features with 64 embedding dimensions, 2 encoder layers, and 2 decoder layers. All configurations maintain *mlp_hidden_times* = 4 across datasets, where *mlp_hidden_times* is a scaling factor that determines the expansion ratio of the hidden layer in the MLP, which is kept consistent with baseline implementations to ensure fair comparison. For stable training and enhanced fusion quality, the complete implementation employs an extended total loss function as an extension of Equation (12):

$$\mathcal{L}_{\text{total}}^{\text{impl}} = \mathcal{L}_T + \mathcal{L}_S + \lambda_{\text{fuse}}\mathcal{L}_{\text{fuse}} + \lambda_{\text{var}}\mathcal{L}_{\text{var}}, \tag{17}$$

where $\mathcal{L}_{\text{var}} = \left(\text{Var}(\hat{x}_t^{\text{fuse}}) - \text{Var}(x_t)\right)^2$ is a variance penalty term that ensures the fused output maintains statistical consistency with the original data. The hyperparameters are configured as $\lambda_{\text{periodic}} = \sqrt{L}\,/\,5$ (used in Equation (10)), $\lambda_{\text{fuse}} = 2.0$ (used in Equation (11)), and $\lambda_{\text{var}} = 0.05$ across all experiments, where $L$ denotes the sequence length. The trend model incorporates SwiGLU activation functions and layer-scale normalization, with an initialization factor $\gamma = 10^{-3}$. In contrast, the seasonality model utilizes GELU activation functions with an optional GELU$^2$ variant.

Both encoder architectures employ identical configurations with AdaLayerNorm for timestep conditioning, FullAttention for self-attention mechanisms, and residual scaling parameters initialized to

$1 \times 10^{-3}$. The trend decoder employs **TrendBlock** module consisting of two sequential 1D convolutions: first transforming from embedding dimension to 3 intermediate channels with kernel size 3 and padding 1, followed by GELU activation, then expanding back to feature dimension with identical convolution parameters. The SwiGLU activation function is defined as:

$$\text{SwiGLU}(x) = W_3\big(\text{SiLU}(xW_1) \odot (xW_2)\big), \tag{18}$$

where $\odot$ denotes element-wise multiplication and $W_1, W_2, W_3$ are projection matrices with hidden dimension calculated as $\left(\frac{2}{3} \times 4 \times d_{\text{model}}\right)$.

The trend decoder incorporates CrossAttention for encoder–decoder interaction, and linear projection layers for adjusting the mean output dimension (embedding to feature dimension). It accumulates TrendBlock outputs across all decoder layers through element-wise summation. The seasonality decoder employs **FourierLayer** module that implements frequency-domain processing corresponding to $\mathcal{D}_S$. The layer-specific feature encoding $\xi_k(x)$ corresponds to the MLP preprocessing with Linear→GELU→Linear→Dropout architecture, while adaptive scaling factors $\mathcal{A}_k$ and learnable phase shifts $\phi_k$ enable periodic pattern alignment. The seasonality decoder utilizes dual AdaLayerNorm layers for enhanced timestep conditioning and processes FourierLayer outputs through standard MLP layers with an expansion factor of 4.

Both architectures utilize **LearnablePositionalEncoding** module for encoder and decoder pathways, with final encoder normalization applied before decoder processing. The trend model applies **OutputAffine** transformations with learnable scale and shift parameters. The trend model includes a combine module (1D convolution with kernel size 1) that aggregates decoder layer outputs from dimension (batch, sequence, *num_decoder_layers*) to (batch, sequence, 1), where *num_decoder_layers* represents the total number of decoder layers. The seasonality model utilizes **OutputScaler** normalization to implement the adaptive scaling mechanism $\mathcal{A}_k$ from the $\mathcal{D}_S$ formulation, with a combine module performing dimensional reduction from embedding to feature space. Additionally, a fusion MLP module integrates both Transformer outputs through Linear→GELU→Linear layers, reducing concatenated features from $2 \times d_{\text{feature}}$ to $d_{\text{model}}$, then to $d_{\text{feature}}$ for final reconstruction.

## C  EXPERIMENTAL SETTINGS

**Metrics.** To evaluate the quality of synthetic time series from multiple perspectives, we employed 11 quantitative metrics. Root mean squared error (RMSE) (Chai & Draxler, 2014) measures the average squared deviation between original and synthetic sequences. The peak signal-to-noise ratio (PSNR) (Huynh-Thu & Ghanbari, 2008) assesses signal quality based on the ratio of the signal power to the noise power. Cosine similarity (COS) (Singhal, 2001) quantifies directional similarity between two sequences. For distributional differences, the maximum mean discrepancy (MMD) (Gretton et al., 2012) is used to evaluate overall distributional divergence. The marginal distribution difference (MDD) measures how well the synthetic data matches original data by comparing the statistical distributions at each time point and for each variable separately. To assess structural properties, autocorrelation-based distance (ACD) (Montero & Vilar, 2014) evaluates similarity in autocorrelation structures, and spectral distance (SD) (Welch, 1967) measures differences in the frequency domain. Kolmogorov–Smirnov distance (KD) (Massey Jr, 1951) compares cumulative distributions, while Euclidean distance (ED) calculates straight-line similarity between sequences. Finally, dynamic time warping (DTW) (Berndt & Clifford, 1994) measures alignment-based similarity, accommodating temporal distortions in the series.

To evaluate the distributional fidelity and statistical consistency of synthetic time-series data, we also employ four additional metrics. The Variational Distance Score (VDS) measures the distributional similarity between original and generated samples based on statistical distance. The Fréchet Deep Distance Score (FDDS) extends the widely used Fréchet Inception Distance to time-series by embedding sequences into a deep feature space and computing the Fréchet distance of their distributions. Both VDS and FDDS metrics are introduced and detailed in (Li et al., 2025). The Discriminative Score (DA) assesses the distinguishability of the original and synthetic data using an auxiliary classifier. The Predictive Score (PD) quantifies the utility of synthetic data for forecasting tasks by measuring prediction errors when models trained on synthetic data are tested on real data. These metrics are originally proposed in (Yoon et al., 2019).

Table 4: Dataset Details

| Dataset | # of Samples | dim | Link |
|---|---|---|---|
| Stocks | 3,662 | 6 | https://finance.yahoo.com/quote/GOOG |
| ETTh | 17,420 | 7 | https://github.com/zhouhaoyi/ETDataset |
| Energy | 19,735 | 28 | https://archive.ics.uci.edu/ml/datasets |

**Datasets.** Table 4 presents the dataset statistics, with all datasets publicly accessible through the provided online links.

## D  LONG TIME SERIES GENERATION FOR WHOLE DATASETS

Table 5 presents the performance comparison between the proposed DDM-TS and baseline methods (TimeVAE, TimeGAN, Diffusion-TS, and PaD-TS) on three datasets—Stock, Energy, and ETTh—across longer sequence lengths (64, 128, and 256). The analysis reveals that DDM-TS consistently outperforms other methods across nearly all datasets and sequence lengths. In particular, it records the lowest values for distribution-based metrics such as RMSE, MMD, and MDD, in most cases. DDM-TS additionally demonstrates superior performance in structural preservation metrics, achieving the highest PSNR and COS scores across all evaluated cases. This result confirms

Table 5: Results on time series generation with longer sequence lengths (64, 128, 256). The best performance is indicated in **bold red**, while the second-best is indicated in **bold blue**.

| Length | Dataset | Method | RMSE | PSNR | COS | MMD | MDD | ACD | SD | KD | ED | DTW |
|---|---|---|---|---|---|---|---|---|---|---|---|---|
| 64 |  | TimeVAE | 0.2769 | 7.4222 | 0.9410 | 0.2133 | 0.0591 | 0.1052 | 0.3128 | 2.2892 | 5.4528 | 41.5299 |
|  |  | TimeGAN | 0.2775 | 7.0184 | 0.9645 | 0.1972 | 0.0569 | 0.0352 | 0.8497 | 3.2732 | 5.1890 | 40.9216 |
|  |  | Diffusion-TS | 0.2224 | 10.2559 | 0.9760 | 0.1677 | 0.0418 | 0.0486 | 0.4063 | 2.4572 | 4.4806 | 33.5574 |
|  |  | PaD-TS | 0.2488 | 9.1141 | 0.9689 | 0.2009 | 0.0721 | 0.0072 | 0.1714 | 1.935 | 5.0131 | 40.1413 |
|  |  | DDM-TS | 0.0587 | 11.6459 | 0.9786 | 0.0091 | 0.0134 | 0.1769 | 0.5184 | 2.4131 | 1.1195 | 7.3721 |
| 128 | Stock | TimeVAE | 0.2787 | 7.2888 | 0.9470 | 0.2290 | 0.0371 | 0.0973 | 0.3723 | 2.4680 | 7.9141 | 90.9010 |
|  |  | TimeGAN | 0.2357 | 6.0640 | 0.9796 | 0.1773 | 0.0943 | 0.2011 | 1.4815 | 3.8376 | 7.1072 | 84.2559 |
|  |  | Diffusion-TS | 0.2112 | 10.3070 | 0.9766 | 0.1613 | 0.0452 | 0.0522 | 0.5285 | 3.3167 | 6.1996 | 66.3955 |
|  |  | PaD-TS | 0.2311 | 9.5780 | 0.9681 | 0.1810 | 0.0155 | 0.0747 | 0.3237 | 3.1424 | 6.6868 | 73.7172 |
|  |  | DDM-TS | 0.0584 | 11.2862 | 0.9832 | 0.0100 | 0.0133 | 0.2114 | 0.5342 | 2.5292 | 1.7101 | 15.6863 |
| 256 |  | TimeVAE | 0.2617 | 7.9162 | 0.9584 | 0.2160 | 0.0324 | 0.0536 | 0.3093 | 2.2044 | 10.7123 | 170.748 |
|  |  | TimeGAN | 0.2604 | 8.0722 | 0.9821 | 0.1710 | 0.0537 | 0.1775 | 0.7482 | 3.1607 | 9.5030 | 146.553 |
|  |  | Diffusion-TS | 0.1827 | 10.8225 | 0.9770 | 0.1343 | 0.0561 | 0.0645 | 0.6924 | 5.5781 | 8.0049 | 114.353 |
|  |  | PaD-TS | 0.2047 | 10.0833 | 0.9668 | 0.1602 | 0.0167 | 0.1075 | 0.2085 | 1.4203 | 8.8394 | 153.782 |
|  |  | DDM-TS | 0.0807 | 13.2015 | 0.9647 | 0.0102 | 0.0135 | 0.2603 | 0.5093 | 2.6971 | 2.5335 | 31.5303 |
| 64 |  | TimeVAE | 0.2444 | 6.6180 | 0.9573 | 0.1226 | 0.0581 | 0.0544 | 0.2117 | 0.8817 | 10.2529 | 81.9925 |
|  |  | TimeGAN | 0.2414 | 6.1689 | 0.9663 | 0.1342 | 0.0248 | 0.1073 | 0.4936 | 3.9693 | 10.7745 | 84.0227 |
|  |  | Diffusion-TS | 0.2502 | 6.1254 | 0.9616 | 0.1347 | 0.0120 | 0.0330 | 0.1642 | 1.1711 | 10.8080 | 83.4565 |
|  |  | PaD-TS | 0.2502 | 6.2282 | 0.9621 | 0.1317 | 0.0067 | 0.0192 | 0.1047 | 0.8082 | 10.6811 | 82.7599 |
|  |  | DDM-TS | 0.1570 | 7.9677 | 0.9795 | 0.0507 | 0.0193 | 0.2086 | 0.2667 | 0.5302 | 6.7205 | 49.5179 |
| 128 | Energy | TimeVAE | 0.2380 | 6.6400 | 0.9588 | 0.1193 | 0.0696 | 0.0565 | 0.3218 | 1.0786 | 14.3788 | 163.224 |
|  |  | TimeGAN | 0.2499 | 6.0654 | 0.9634 | 0.1355 | 0.0203 | 0.1113 | 0.2135 | 0.7700 | 15.3619 | 173.356 |
|  |  | Diffusion-TS | 0.2495 | 6.0919 | 0.9618 | 0.1338 | 0.0098 | 0.0287 | 0.1155 | 0.4952 | 15.2852 | 169.340 |
|  |  | PaD-TS | 0.2490 | 6.1612 | 0.9620 | 0.1318 | 0.0060 | 0.0178 | 0.1022 | 0.7535 | 15.1672 | 167.569 |
|  |  | DDM-TS | 0.1640 | 8.3527 | 0.9780 | 0.0546 | 0.0189 | 0.2121 | 0.2637 | 0.5339 | 9.9007 | 103.551 |
| 256 |  | TimeVAE | 0.2503 | 6.0616 | 0.9553 | 0.1314 | 0.0770 | 0.0754 | 0.3028 | 1.0194 | 21.5496 | 344.110 |
|  |  | TimeGAN | 0.2682 | 6.0658 | 0.9604 | 0.1317 | 0.0326 | 0.1524 | 0.4796 | 1.1020 | 21.5635 | 339.411 |
|  |  | Diffusion-TS | 0.2481 | 6.0616 | 0.9620 | 0.1334 | 0.0097 | 0.0168 | 0.0840 | 0.4419 | 21.6401 | 327.580 |
|  |  | PaD-TS | 0.2478 | 6.1638 | 0.9623 | 0.1309 | 0.0065 | 0.0256 | 0.0839 | 0.4880 | 21.4075 | 332.324 |
|  |  | DDM-TS | 0.1685 | 7.7891 | 0.9770 | 0.0574 | 0.0181 | 0.2871 | 0.2662 | 0.5199 | 14.3998 | 208.329 |
| 64 |  | TimeVAE | 0.2031 | 7.8998 | 0.9837 | 0.0901 | 0.0380 | 0.0693 | 0.2512 | 1.2803 | 4.4058 | 32.3170 |
|  |  | TimeGAN | 0.1700 | 8.2017 | 0.9850 | 0.0797 | 0.0295 | 0.0688 | 0.6115 | 1.5114 | 4.1018 | 31.2629 |
|  |  | Diffusion-TS | 0.1800 | 8.5593 | 0.9837 | 0.0782 | 0.0126 | 0.0224 | 0.1899 | 0.8909 | 4.0932 | 29.4189 |
|  |  | PaD-TS | 0.1779 | 8.9026 | 0.9838 | 0.0728 | 0.0192 | 0.0375 | 0.2194 | 1.5410 | 3.9402 | 29.7993 |
|  |  | DDM-TS | 0.1343 | 10.7585 | 0.9878 | 0.0398 | 0.0201 | 0.0904 | 0.3793 | 1.3978 | 2.9428 | 21.2494 |
| 128 | ETTh | TimeVAE | 0.2026 | 7.7633 | 0.9837 | 0.0911 | 0.0454 | 0.0674 | 0.1450 | 0.6434 | 6.2951 | 64.2599 |
|  |  | TimeGAN | 0.1700 | 8.2106 | 0.9818 | 0.0812 | 0.0247 | 0.1666 | 0.4338 | 1.3773 | 5.9499 | 61.2670 |
|  |  | Diffusion-TS | 0.1818 | 8.6818 | 0.9829 | 0.0755 | 0.0214 | 0.0399 | 0.2172 | 1.4075 | 5.6945 | 59.3000 |
|  |  | PaD-TS | 0.1781 | 8.8416 | 0.9834 | 0.0732 | 0.0183 | 0.0521 | 0.2155 | 1.5382 | 5.5960 | 56.5181 |
|  |  | DDM-TS | 0.1355 | 10.046 | 0.9875 | 0.0418 | 0.0214 | 0.1123 | 0.4229 | 1.4193 | 4.2801 | 42.3246 |
| 256 |  | TimeVAE | 0.2005 | 7.7685 | 0.9837 | 0.0894 | 0.0436 | 0.0500 | 0.2031 | 0.4556 | 8.8563 | 122.784 |
|  |  | TimeGAN | 0.2084 | 7.6627 | 0.9785 | 0.0843 | 0.0617 | 0.1925 | 0.7292 | 2.3165 | 8.5595 | 133.866 |
|  |  | Diffusion-TS | 0.1810 | 8.7111 | 0.9823 | 0.0740 | 0.0259 | 0.0612 | 0.2392 | 1.7372 | 7.9978 | 113.288 |
|  |  | PaD-TS | 0.1790 | 8.7011 | 0.9832 | 0.0740 | 0.0175 | 0.0666 | 0.1809 | 1.1761 | 7.9989 | 109.892 |
|  |  | DDM-TS | 0.1407 | 10.4836 | 0.9869 | 0.0443 | 0.0224 | 0.1358 | 0.5016 | 1.3757 | 6.2493 | 84.7581 |

Table 6: Results of population-level properties obtained on multiple time-series datasets with longer sequence lengths (64, 128, 256). The best performance is indicated in **bold**.

| Length | Dataset | Method | VDS | FDDS | DA | PD | Length | Dataset | Method | VDS | FDDS | DA | PD |
|---|---|---|---|---|---|---|---|---|---|---|---|---|---|
| 64 | Stock | TimeVAE | 0.1269 | 0.3713 | 0.3633 | 0.1396 | 64 | Energy | TimeVAE | 0.2656 | 0.2240 | 0.4997 | 0.2691 |
| | | TimeGAN | 0.1074 | 0.1918 | 0.2011 | 0.0387 | | | TimeGAN | 0.0220 | 0.5519 | 0.4819 | 0.3076 |
| | | Diffusion-TS | 0.0381 | 0.0555 | 0.0924 | 0.0369 | | | Diffusion-TS | 0.0063 | 0.0785 | 0.2004 | **0.2509** |
| | | PaD-TS | **0.0011** | **0.0167** | **0.0699** | **0.0365** | | | PaD-TS | **0.0020** | **0.0222** | **0.1060** | 0.2519 |
| | | DDM-TS | 0.0030 | 0.0304 | 0.0747 | 0.0369 | | | DDM-TS | 0.0070 | 0.2187 | 0.3612 | 0.2529 |
| 128 | Stock | TimeVAE | 0.0847 | 0.4260 | 0.3089 | 0.1362 | 128 | Energy | TimeVAE | 0.3269 | 0.1798 | 0.4999 | 0.2636 |
| | | TimeGAN | 0.2566 | 0.2109 | 0.2410 | 0.0462 | | | TimeGAN | 0.0160 | 0.4740 | 0.3529 | 0.3028 |
| | | Diffusion-TS | 0.0477 | 0.0384 | 0.1381 | 0.0370 | | | Diffusion-TS | 0.0031 | 0.0295 | 0.2503 | **0.2503** |
| | | PaD-TS | **0.0043** | 0.0282 | **0.0417** | **0.0362** | | | PaD-TS | **0.0016** | **0.0169** | **0.1097** | 0.2515 |
| | | DDM-TS | 0.0050 | **0.0103** | 0.0743 | 0.0368 | | | DDM-TS | 0.0088 | 0.3209 | 0.2608 | 0.2542 |
| 256 | Stock | TimeVAE | 0.0338 | 0.0619 | 0.2602 | 0.0755 | 256 | Energy | TimeVAE | 0.4229 | 0.1183 | 0.5000 | 0.2546 |
| | | TimeGAN | 0.1041 | 0.3348 | 0.2173 | 0.0362 | | | TimeGAN | 0.0441 | 1.3821 | 0.4586 | 0.2791 |
| | | Diffusion-TS | 0.0920 | 0.0768 | 0.1615 | 0.0362 | | | Diffusion-TS | 0.0026 | **0.0064** | 0.3534 | **0.2496** |
| | | PaD-TS | 0.0089 | 0.0386 | **0.0840** | **0.0353** | | | PaD-TS | **0.0017** | 0.0167 | **0.3483** | 0.2505 |
| | | DDM-TS | **0.0082** | **0.0123** | 0.0870 | 0.0357 | | | DDM-TS | 0.0107 | 0.3169 | 0.4029 | 0.2947 |
| 64 | ETTh | TimeVAE | 0.1574 | 0.3021 | 0.4310 | 0.1230 | | | | | | | |
| | | TimeGAN | 0.0352 | 1.2952 | 0.3620 | 0.1558 | | | | | | | |
| | | Diffusion-TS | 0.0062 | 0.0866 | 0.0864 | **0.1200** | | | | | | | |
| | | PaD-TS | 0.0161 | 0.1706 | 0.1671 | 0.1229 | | | | | | | |
| | | DDM-TS | **0.0057** | **0.0316** | **0.0755** | 0.1216 | | | | | | | |
| 128 | ETTh | TimeVAE | 0.1983 | 0.1984 | 0.4384 | 0.1250 | | | | | | | |
| | | TimeGAN | 0.0247 | 1.1461 | 0.3563 | 0.1553 | | | | | | | |
| | | Diffusion-TS | 0.0195 | 0.1429 | 0.1585 | 0.1223 | | | | | | | |
| | | PaD-TS | 0.0138 | 0.1467 | 0.2310 | **0.1210** | | | | | | | |
| | | DDM-TS | **0.0066** | **0.0405** | **0.0921** | 0.1244 | | | | | | | |
| 256 | ETTh | TimeVAE | 0.1755 | 0.1394 | 0.4164 | 0.1235 | | | | | | | |
| | | TimeGAN | 0.1381 | 1.5912 | 0.4762 | 0.1595 | | | | | | | |
| | | Diffusion-TS | 0.0262 | 0.1902 | 0.2108 | 0.1223 | | | | | | | |
| | | PaD-TS | 0.0134 | 0.2038 | 0.2052 | 0.1265 | | | | | | | |
| | | DDM-TS | **0.0059** | **0.0355** | **0.0909** | **0.1191** | | | | | | | |

that the generated time series effectively maintains overall similarity to the original data. The autocorrelation and frequency-domain consistency metrics (ACD, SD, KD) consistently exhibit poor performance across all sequence lengths, indicating difficulty in preserving sequential dependencies and distributional properties of the original time series. For distance-based metrics such as ED and DTW, DDM-TS achieves the lowest error values compared to competing methods, demonstrating that the generated sequences maintain enhanced temporal alignment and distance consistency with the original data patterns. In conclusion, DDM-TS demonstrates enhanced reconstruction quality but faces trade-offs in maintaining certain temporal correlation patterns in the original data.

Table 6 presents comprehensive results for population-level property preservation across multiple datasets (Stock, Energy, ETTh) and longer sequence lengths (64, 128, 256), evaluating four key metrics: VDS, FDDS, DA, and PD. On the Stock dataset, DDM-TS maintains competitive performance across all sequence lengths, achieving results comparable to PaD-TS. The ETTh dataset demonstrates the strongest performance, with DDM-TS frequently achieving the best results across most metrics and sequence lengths. However, the Energy dataset reveals weaker performance compared to the Stock and ETTh datasets. These results indicate that DDM-TS has limitations in preserving certain population-level characteristics on this dataset. Overall, DDM-TS demonstrates mixed performance, with strong results on the Stock and ETTh datasets that are comparable to PaD-TS, but exhibits limitations on the Energy dataset. This result highlights the dataset-dependent effectiveness of DDM-TS and suggests that it may struggle with certain temporal characteristics that vary across datasets.

# E    DATA DISTRIBUTION ANALYSIS FOR WHOLE DATASETS

Figure 6 presents t-SNE visualizations and kernel density distributions comparing the distributional fidelity and coverage of synthetic data generation across different methods on various datasets. Analysis of the Stock dataset results is provided in the **data distribution analysis** in Section 4.1.

In the ETTh dataset, TimeVAE demonstrates the most restricted generation capability with poor distributional matching. TimeGAN displays patterns in both t-SNE visualization and density estimation that deviate from the original distribution characteristics. DDM-TS achieves distributional coverage comparable to Diffusion-TS and PaD-TS in the t-SNE visualization. However, the ETTh results reveal limitations, as the density estimation plot shows incomplete coverage in certain regions

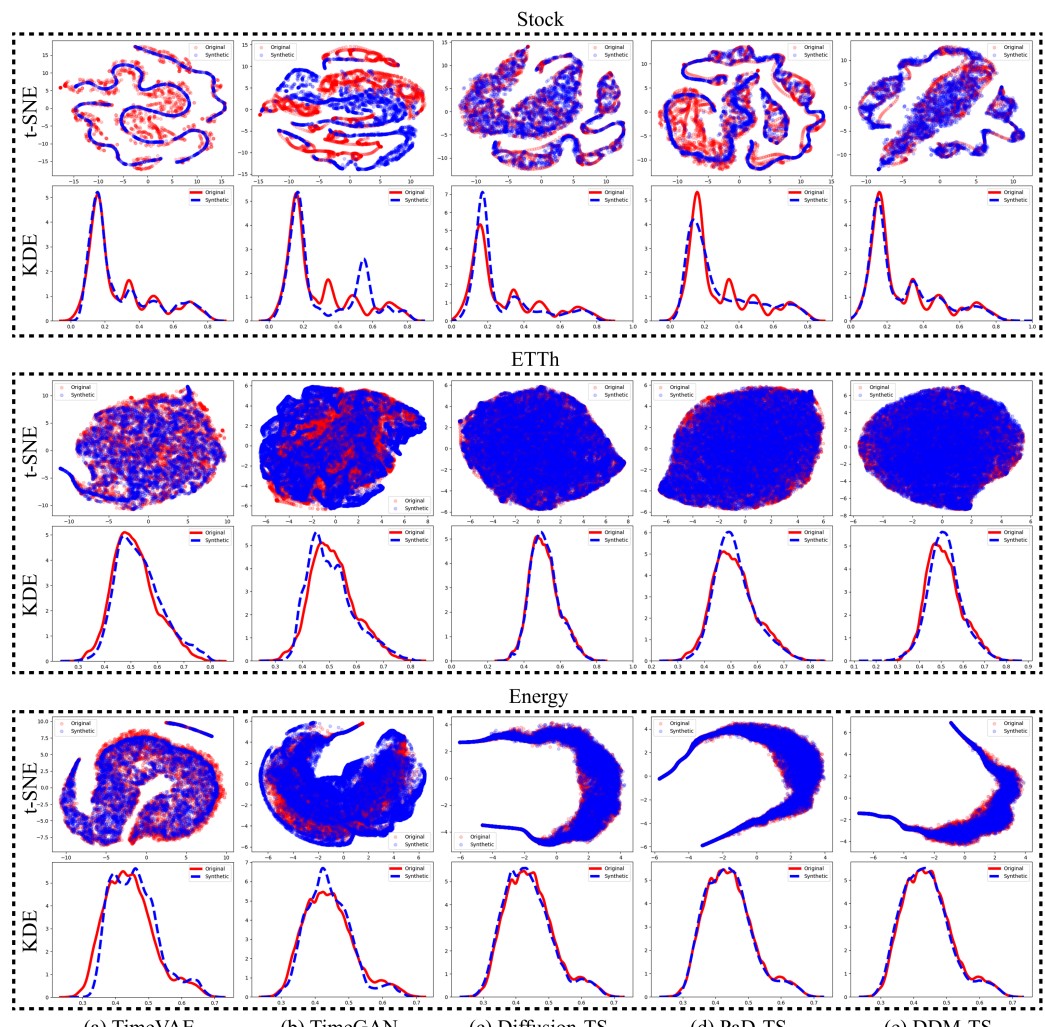

Figure 6: Comparison of data distributions from different time-series generation methods across various datasets

compared to Diffusion-TS. These performance variations suggest that the effectiveness of DDM-TS depends on the underlying temporal characteristics and the inherent complexity of the target dataset.

In the Energy dataset, TimeVAE exhibits restricted generation capabilities, with synthetic points concentrated in limited areas and density estimation deviating from the original data. TimeGAN exhibits better coverage than TimeVAE, but it displays clustering patterns that deviate from the original distribution in t-SNE visualizations. Diffusion-TS demonstrates moderate performance with density estimation, showing minimal deviation from the original data. DDM-TS achieves distributional coverage comparable to both Diffusion-TS and PaD-TS in the t-SNE visualization. The corresponding density estimation plot demonstrates excellent alignment between the original and synthetic distributions.

These distributional analysis results demonstrate that while DDM-TS maintains competitive performance across various datasets, its effectiveness in achieving complete distributional coverage varies significantly depending on the complexity and temporal structure of the target dataset.

# F  USAGE OF LARGE LANGUAGE MODELS

We utilized large language models (LLMs) to assist with refining and polishing the language throughout manuscript preparation. Additionally, LLMs were employed for literature discovery to identify relevant prior work in diffusion models and time series generation research.

