# OpenReview forum: "DDM-TS: Decoupled Diffusion Models for Time-Series Generation with Explicit Trend–Seasonality Decomposition"
_ICLR.cc/2026/Conference — ICLR 2026 Conference Withdrawn Submission_

### Official Review · Reviewer_Y7Lw · 2025-10-27

**Soundness:** 3
**Presentation:** 3
**Contribution:** 2
**Rating:** 6
**Confidence:** 4

**Summary:**

This paper proposes DDM-TS a decoupled diffusion framework that aims to learn trend and seasonality components from STL decomposition for unconditional time series generation. Empirical results demonstrated strong performance in generation quality through many evaluation metrics.

**Strengths:**

1. The paper is well written and clearly presents the questions, proposed methods, and analysis.
2. As shown in Tables 1 and 2, DDM-TS shows strong performance in selected datasets, especially in the low-dimensional dataset.

**Weaknesses:**

1. As shown in Table 2, DDM-TS has demonstrated strong performance in low-dimensional datasets like (Stock and ETTh). It seems the performance degrades with the Energy dataset, which is inconsistent with Table 1.
2. Due to randomness in generation, it is common to perform DA and PD evaluation multiple times with the standard deviation reported. It will be more compelling if the authors report the STD as commonly used.
3. Only three datasets are evaluated throughout the paper. The proposed methods could be more compelling if more diverse datasets were used in evaluation.
4. Minor: In Figure 2, it will be great to write the name of the dataset in the description.

**Questions:**

1. As written in W1, do the authors have any suggestion why DDM-TS’s performance degrade in high-dimensional datasets?
2. As commonly used in diffusion-based generation models, they usually operate in the raw value space. How does DDM-TS perform if STL decomposition is removed? It will be great to include this experiment in the Ablation study (Table 3).

---

### Official Review · Reviewer_cAYy · 2025-10-30

**Soundness:** 2
**Presentation:** 3
**Contribution:** 2
**Rating:** 4
**Confidence:** 4

**Summary:**

This paper proposes DDM-TS, a novel decoupled diffusion architecture specifically designed for time-series generation. The core mechanism involves a pre-processing step using STL decomposition to separate the long-term trend and seasonal components. Subsequently, two independent Denoising Diffusion Probabilistic Models (DDPMs) are trained—one dedicated to each component. A final fusion stage then reconstructs the full signal. This architectural separation is motivated by the aim to mitigate frequency interference (or gradient conflict across different scales) that typically hinders a single diffusion network. Empirical evaluation on standard benchmarks demonstrates improved reconstruction accuracy (lower RMSE and higher PSNR) when compared to prior time-series diffusion models like Diffusion-TS and TimeGrad, though this gain is reported to come at the cost of reduced spectral fidelity.

**Strengths:**

- The decoupled architecture is soundly implemented, and the underlying rationale of frequency disentanglement is theoretically reasonable for addressing gradient conflicts in multi-scale time series.

- The authors conduct systematic experiments that successfully demonstrate strong reconstruction accuracy ($\downarrow$ RMSE, $\uparrow$ PSNR) compared to existing diffusion models.

- The work represents sound engineering and provides a practical, easily implementable solution well-suited for practitioners in time-series analysis.

**Weaknesses:**

- The reliance on additive decomposition fundamentally oversimplifies real-world temporal dynamics. In most natural and financial processes, the coupling between trend and seasonality is multiplicative, where the magnitude of seasonality scales with the overall trend level.

- The conceptual novelty is limited. The work is primarily architectural and demonstrates a practical engineering workaround rather than introducing new, deep theoretical, generative knowledge about diffusion models.

- There is an ambiguity in the gating/fusion mechanism that needs clarification: the parameter $\alpha_T(t)$ appears both in the trend reconstruction term, $P_{\text{trend}}(\alpha_T(t))$ (Eq. 2), and in the final fusion equation, $\hat{x}_t = \alpha_T(t)\hat{T}_t + \alpha_S(t)\hat{S}_t$ (Eq. 8). Sharing a single parameter for both component decoding and component fusion could potentially re-introduce gradient interference between the decoder and the final reconstruction stage; conversely, separating them would invalidate the current notational consistency.

**Questions:**

- In Section 3.1 and the discussion concerning Figure 4, the authors acknowledge that the purely additive STL decomposition may be inadequate for capturing amplitude variation across time, particularly in datasets where the seasonal component’s magnitude is clearly dependent on the overall trend level (i.e., when volatility or energy scales with the level). Given this well-known limitation, have the authors considered a multiplicative or log-domain formulation that better handles coupled dynamics, such as: $\log x_t = \log T_t + \log(1 + S_t)$?

- The current empirical evaluation appears to be limited exclusively to datasets (ETTh, Energy, Stocks) that naturally exhibit clear periodic or trend–seasonal structures. How would DDM-TS perform, or how would the entire STL-based framework behave, on non-seasonal, event-driven, or highly dynamic datasets, e.g., MuJoCo (physical control signals), CMAPSS (engine degradation signals), EEG/fMRI signals (neurological data)? In such domains, the fundamental assumption of a separable trend and seasonality is often ill-defined or irrelevant. Providing results on such benchmarks is crucial to demonstrate the model's generalizability beyond structured seasonal data.

---

### Official Review · Reviewer_Xw5Z · 2025-11-01

**Soundness:** 2
**Presentation:** 2
**Contribution:** 2
**Rating:** 2
**Confidence:** 4

**Summary:**

This paper introduces DDM-TS, a decoupled diffusion architecture for unconditional time-series generation, explicitly separating trend and seasonality through STL decomposition. Each component is modeled by an independent diffusion process with specialized encoders and decoders (trend modeling emphasizes smoothness), while seasonality modeling employs FFT-based frequency constraints. A gate-based fusion module adaptively combines both components.

**Strengths:**

- Good clarity and well-organized writing.
- Clear motivation: addresses the interference between trend and seasonality in unified diffusion frameworks
- Well-designed decoupled architecture with STL decomposition and component-specific diffusion & adaptive gating mechanism for dynamic trend–seasonality fusion
- Strong experimental performance on multiple benchmarks
- Thorough ablation study and interpretability analysis

**Weaknesses:**

1. Lack of novelty
- While the decoupling of diffusion processes is conceptually interesting, similar decomposition principles (e.g., Diffusion-TS, FALDA) already exist. The paper primarily combines these existing ideas rather than introducing a fundamentally new generative mechanism.
- The STL decomposition and separate DDPMs are straightforward extensions; the innovation lies mostly in integration, not in the generative process itself.

2. Complexity vs. performance trade-off:
- The architecture introduces two diffusion models, each with encoder-decoder stacks, plus a fusion module. This increases computational cost, but no runtime or efficiency comparison is provided.

3. Ablation results lack deeper analysis
- The ablation shows degradation when removing either module but does not discuss fusion parameter sensitivity or training stability—key concerns for multi-path diffusion systems. It remains unclear how the model balances learning dynamics between trend and seasonality during joint optimization.

4. Dependent on metrics
- The performance ranking of DDM-TS varies drastically across different metric categories (e.g., strong in reconstruction but weak in distributional and spectral consistency).

5. Limited dataset diversity
- The experiments are conducted on only three datasets (Stock, Energy, ETTh), all of which are well-known and relatively small-scale.

**Questions:**

Please refer to the weakness above

---

### Official Review · Reviewer_1Zho · 2025-11-03

**Soundness:** 1
**Presentation:** 2
**Contribution:** 1
**Rating:** 2
**Confidence:** 4

**Summary:**

The paper proposes DDM-TS, a “decoupled” diffusion framework for unconditional multivariate time-series generation that separately models trend and seasonality components using STL decomposition and independent diffusion models, followed by an adaptive gating fusion. While the authors claim state-of-the-art performance—highlighting a 33.8% average improvement in RMSE and PSNR—the work suffers from severe conceptual, methodological, and experimental flaws.

**Strengths:**

1. The experimental evaluation is unusually broad, covering multiple datasets, sequence lengths, and over a dozen metrics, which shows commendable effort in empirical coverage, despite flawed interpretation.

2. The qualitative visualizations (e.g., t-SNE, waveform plots) are clearly presented and do reveal some practical utility in capturing macro-level patterns, especially for financial time series.

**Weaknesses:**

1. The core claim that unified diffusion models “suffer from inherent component interference”, is asserted without proof. No formal analysis (e.g., gradient conflict, mutual information leakage) justifies decoupling. STL decomposition is heuristic and non-identifiable; treating its outputs as ground-truth components for diffusion training is statistically unsound.

2. The work is essentially a minor variant of Diffusion-TS (Yuan & Qiao, 2024b), which already uses trend–seasonality decomposition. Claiming “decoupling” as a novelty is misleading. Diffusion-TS also processes components separately. DDM-TS merely adds gating and per-component decoder designs that resemble prior work in seasonal forecasting (e.g., SSDNet, FALDA).

3. While RMSE/PSNR improvements are emphasized, DDM-TS consistently underperforms in autocorrelation (ACD), spectral distance (SD), and diversity metrics (KD), critical for time-series fidelity. These failures are downplayed as “trade-offs,” not weaknesses, which is scientifically dishonest.

4. The trend decoder uses an ad-hoc smoothness filter with learnable thresholds (Eq. 6), but there’s no ablation showing its necessity over standard diffusion backbones. Similarly, the FFT-based seasonality module adds complexity without justification. Why not use standard positional encodings or temporal convolutions?

5. STL decomposition requires the entire time series to estimate trend and seasonality. Yet the method is applied to unconditional generation, implying that clean STL components must be known at training time.

**Questions:**

1. Please address Weakness #1: Provide theoretical or empirical evidence that “component interference” genuinely exists in standard diffusion models for time series.

2. Regarding Weakness #2: Clarify how DDM-TS fundamentally differs from Diffusion-TS (Yuan & Qiao, 2024b) beyond superficial architectural changes.

3. For Weakness #3: Why are poor ACD/SD scores acceptable? How can a method claim “structural integrity” while failing basic autocorrelation tests?

---

### Note · Authors · 2025-12-01

**Comment:**

To all the reviewers, We sincerely appreciate you for your constructive comments on our paper. After thorough consideration, we have decided to withdraw our submission from ICLR 2026. We will apply your valuable comments to our next submission. Best regards, The authors

**Withdrawal Confirmation:**

I have read and agree with the venue's withdrawal policy on behalf of myself and my co-authors.